# Phenotypic and Genotypic Characterization of Macrolide, Lincosamide and Streptogramin B Resistance among Clinical Methicillin-Resistant *S**taphylococcus aureus* Isolates in Chile

**DOI:** 10.3390/antibiotics11081000

**Published:** 2022-07-25

**Authors:** Mario Quezada-Aguiluz, Alejandro Aguayo-Reyes, Cinthia Carrasco, Daniela Mejías, Pamela Saavedra, Sergio Mella-Montecinos, Andrés Opazo-Capurro, Helia Bello-Toledo, José M. Munita, Juan C. Hormazábal, Gerardo González-Rocha

**Affiliations:** 1Laboratorio de Investigación en Agentes Antibacterianos, Departamento de Microbiología, Facultad de Ciencias Biológicas, Universidad de Concepción, Concepción 4070386, Chile; marioquezada@udec.cl (M.Q.-A.); alejandroaguayo@udec.cl (A.A.-R.); cinthiacarrasco@udec.cl (C.C.); dmejias@udec.cl (D.M.); pame.eudec@gmail.com (P.S.); hbello@udec.cl (H.B.-T.); 2Departamento de Medicina Interna, Facultad de Medicina, Universidad de Concepción, Concepción 4070386, Chile; pignatio@outlook.com; 3Millennium Initiative for Collaborative Research on Bacterial Resistance (MICROB-R), Santiago 3580000, Chile; munita.jm@gmail.com; 4Centro Regional de Telemedicina y Telesalud del Biobío (CRT Biobío), Concepción 4030000, Chile; 5Unidad de Medicina Interna, Hospital Traumatológico de Concepción, Concepción 4030000, Chile; 6Unidad de Infectología, Hospital Regional “Dr. Guillermo Grant B.”, Concepción 4030000, Chile; 7Genomics and Resistant Microbes (GeRM) Group, Facultad de Medicina, Clínica Alemana Universidad del Desarrollo, Santiago 7550000, Chile; 8Subdepartamento de Enfermedades Infecciosas, Instituto de Salud Pública de Chile, Santiago 7780050, Chile; jchormazabal@ispch.cl

**Keywords:** MRSA, MLS_B_ phenotype, antibiotic-resistant

## Abstract

Macrolides, lincosamides, and type B streptogramins (MLS_B_) are important therapeutic options to treat methicillin-resistant *Staphylococcus aureus* (MRSA) infections; however, resistance to these antibiotics has been emerging. In Chile, data on the MLS_B_ resistance phenotypes are scarce in both community-(CA) and hospital-acquired (HA) MRSA isolates. Antimicrobial susceptibility to MLS_B_ was determined for sixty-eight non-repetitive isolates of each HA-(32) and CA-MRSA (36). Detection of SCC*mec* elements, *ermA*, *ermB*, *ermC*, and *msrA* genes was performed by PCR. The predominant clones were SCC*mec* I-ST5 (HA-MRSA) and type IVc-ST8 (CA-MRSA). Most of the HA-MRSA isolates (97%) showed resistance to clindamycin, erythromycin, azithromycin, and clarithromycin. Among CA-MRSA isolates, 28% were resistant to erythromycin, azithromycin, and 25% to clarithromycin. All isolates were susceptible to linezolid, vancomycin, daptomycin and trimethoprim/sulfamethoxazole, and over 97% to rifampicin. The *ermA* gene was amplified in 88% of HA-MRSA and 17% of CA-MRSA isolates (*p* < 0.001). The *ermC* gene was detected in 6% of HA-SARM and none of CA-SARM isolates, whereas the *msrA* gene was only amplified in 22% of CA-MRSA (*p* < 0.005). Our results demonstrate the prevalence of the cMLSB resistance phenotype in all HA-MRSA isolates in Chile, with the *ermA* being the predominant gene identified among these isolates.

## 1. Introduction

Methicillin-resistant *Staphylococcus aureus* (MRSA) is an important pathogen involved in both human and animal infections [1,2]. Although MRSA was initially described as producing healthcare-associated infections (HA-MRSA), the appearance of community-associated MRSA infections (CA-MRSA) has been documented since the 1990s [3]. MRSA has shown a remarkable ability to develop resistance to a myriad of antibiotics, as well as to different disinfectants and heavy metals [4]. Vancomycin (VAN), a member of the glycopeptides, has been used as an important option to treat MRSA infections [5]. However, the risk of dissemination of vancomycin-resistant or non-fully susceptible strains suggests that this antibiotic should be used sparingly [6]. For this reason, macrolides (erythromycin [ERY]), lincosamides (clindamycin [CLI]), and streptogramins B (MLS_B_) have emerged as important therapeutic options to tackle CA-MRSA infections [7,8]. However, the increased use of these antimicrobials has favored the emergence of resistance to these drugs [9,10,11]. To date, there are three main MLS_B_ resistance mechanisms described: i) changes in the ribosomal target site, which confers cross-resistance to the entire MLS_B_ group [12]. This mechanism is conferred by ribosomal mutations or methylation of the 23S rRNA target site, which are mediated by the *erm* genes (mainly *erm*A, *erm*B, and *erm*C) [13,14]. Another mechanism corresponds to ii) an efflux-pump encoded by *msrA*, which can drive out 14- and 15-membered macrolides and streptogramin B, producing the MS_B_ phenotype [15]. Finally, another mechanism iii) relies on drug inactivation and it only confers resistance to lincosamides due to an enzyme encoded by the *lnu* gene [11].

Significantly, the MLS_B_ phenotype can be either constitutive (cMLS_B_) or inducible (iMLS_B_) [9]. Specifically, CLI, which is the MLS_B_ agent used for the treatment of *S. aureus* infections, is a weak MLS_B_-resistance inducer and may lead to treatment failure due to false susceptibility results displayed in in vitro antimicrobial susceptibility tests [16]. Therefore, it is necessary to perform the CLI susceptibility test in the presence of a strong inducer, such as ERY [12]. Another key point is that antibiotic resistance genes that mediate the MLS_B_-resistance phenotype are found in mobile-genetic elements (MGEs) and, in consequence, may be horizontally transferred to susceptible strains [17]. In Latin America, the resistance rates to MLS_B_ antibiotics have been reported to be 74% and 81% to ERY and CLI, respectively, among HA-MRSA isolates [10].

In Chile, *S. aureus* is one of the main etiological agents in health care-associated infections (HAIs) [18]. Specifically, it is the main cause of surgical wound infections (27%), and the second cause of pneumonia associated to invasive mechanical ventilation (21%). Likewise, it is involved in bloodstream infections (18%) and infections of the central nervous system (18%) [18]. Despite these data, the MLS_B_-resistance phenotype among HA- and CA-MRSA is still unknown among Chilean isolates. Therefore, the aim of our study was to detect and characterize the MLS_B_- and MS_B_-resistance phenotypes among HA-MRSA and CA-MRSA isolates collected between 2007 and 2017 from the *S. aureus* surveillance program of the National Institute of Public Health of Chile (ISP).

## 2. Results

### 2.1. Molecular Characterization of MRSA Isolates

All HA-MRSA (32) and CA-MRSA (36) isolates were resistant to FOX and *mecA* positive. For HA-MRSA, the Staphylococcal Cassette Chromosome *mec* (SSC*mec*) analysis revealed the presence of the Type I and Type II elements in 27 (84.4%) and 5 (15.6%) isolates, respectively. In addition, in all isolates classified as HA-MRSA, the absence of the *pvl* gene was confirmed. On the other hand, in all CA-MRSA (36), the *pvl* gene and the type IV SSC*mec* cassette were detected. Of these, 24 (66.7%) harbored the cassette subtype SSC*mec* IVc, whereas 11 (30.5%), and 1 (2.8%) amplified for the subtypes IVa and IVb, respectively; therefore, they were confirmed as CA-MRSA.

The MLST analyses of HA-MRSA showed that 27 (84.4%) isolates belonged to ST5 and 5 (15.6%) to ST105, whereas most CA-MRSA isolates belonged to the ST8 (27/36) (Table 1).

### 2.2. Antimicrobial Susceptibility Testing 

The antibiotic resistance profiles were determined for both HA-MRSA and CA-MRSA isolates (Table 2). All isolates (32) of HA-MRSA were resistant to macrolides and to CLI. Moreover, 2 isolates (2/32) (6.3%) were also resistant to CHL and 1 isolate (1/32) (3.1%) to RIF. In the case of CA-MRSA, 9 isolates (9/36) (25%) were resistant to ERY, AZM and CLR, and one isolate was resistant to ERY and AZT (2.8%), but all were susceptible to CLI, CHL, and RIF (Table 2). All HA-MRSA, and CA-MRSA isolates were susceptible to LZD, VAN, DAP, and SXT (Table 3). Furthermore, the iMLS mechanism was detected in none of the two groups of MRSA isolates.

The HA-MRSA group showed more extended resistance profiles than CA-MRSA. Among the HA-MRSA, the most prevalent resistance profile was CLI-ERY-AZM-CLR, with 90.6% of isolates. On the other hand, in the CA-MRSA group, the most prevalent antibiotic resistance profile was ERY, AZM, and CLR, with 25% of isolates.

### 2.3. Prevalence of msrA and erm Genes

The *ermA* gene was amplified in 28 (87.5%) HA-MRSA isolates compared with 6 (16.7%) in CA-MRSA (*p* < 0.001). Additionally, the *ermC* gene was found in 2 (6.3%) of HA-MRSA and in none of CA-MRSA isolates (*p* > 0.05), and the *ermB* gene was detected in none of the isolates. On the other hand, *msrA* was detected in 11 (30.6%) of the CA-MRSA isolates, but in none of the HA-MRSA (*p* < 0.005) (Table 4).

## 3. Discussion

In recent years, we have observed an increased resistance to antibiotics, especially in those used for the treatment of serious infections associated with health care. MLS_B_ group are antibiotics commonly used to treat skin and soft tissue infections caused by CA-MRSA [11]. The present study reports percentages of resistance to antibiotics in the MLS_B_ group ≥90% in HA-MRSA. This finding agrees with the results of previous studies carried out with strains collected in Chile [10,19]. Besides, 20% of strains of CA-MRSA were resistant to MLS_B_ group. These results show lower rates of resistance to these antibiotics in comparison to the official reports of the National Institute of Public Health of Chile (20% v/s 29%, respectively). On the other hand, our results showed higher values than previous reports that included strains isolated in Latin America, among both HA-MRSA (81% for ERY and 74% for CLI) and CA-MRSA [9,10,20,21,22,23,24]. 

Among the isolates included in this work, the predominant phenotype was the cMLS_B_ phenotype. Molecular characterization of 68 MLSB-resistant MRSA revealed that among HA-MRSA, 87.5% were positive for *ermA*. However, in the CA-MRSA strains, 16.7% were positive for *ermA*, 6.3% for *ermC*, and 30.6% for *msrA*. The main mechanism of resistance to macrolides in CA-MRSA is mediated by the presence of the *msrA* gene, which results agree with previously published data [25]. 

Our results are in agreement with previous reports about the predominance of SCC*mec* type I-ST5 in HA-MRSA in Chile with classic resistance profiles of the Chilean/Cordobes clone that has a marked presence in hospitals of our country [10,26], and isolates of type IV-ST8 in CA-MRSA in Latin America, related to the USA-300 clone [10,19]. On the other hand, the dichotomy regarding the presence of MLS_B_ or MS_B_ resistance among HA-MRSA isolates highlights compared with CA-MRSA (97% vs approximately 25%, reaching statistical significance, *p* < 0.005). However, it is important to emphasize that these findings, which are consistent with the classic concept that hospital isolates of MRSA are multi-resistant and the community-based multi-susceptible and only resistant to β-lactams, should be monitored, since 20% of the isolates of CA-MRSA were resistant to antibiotics in this group, that is, 1 over 5 isolates were not widely susceptible. Accordingly, it is important to perform the proper laboratory detection of these phenotypes to analyze these isolates, since if the criterion of resistance to methicillin and broad susceptibility is the method of choice, other families, including those of the MLS_B_ group, could obtain biased results.

All the strains analyzed are susceptible to VAN, LZD, DAP, and SXT, keeping these antibiotics as an alternative treatment within the therapeutic arsenal available in Chile, which is consistent with previous reports [10,18].

In summary, despite the higher frequency of the cMLS_B_ phenotype than iMLS_B_ in this study, we recommend performing the D test to identify clindamycin-induced resistance and guide therapeutic procedures in both HA-MRSA and CA-MRSA. Likewise, it is not recommended ruling out the submission of suspected CA-MRSA strains in surveillance programs based exclusively on the criterion of resistance only to β-lactams.

## 4. Materials and Methods

### 4.1. MRSA Isolates

Thirty-two non-repetitive HA-MRSA isolates recovered from eight Chilean cities between 2007 and 2017 (Table 5), and thirty-six CA-MRSA isolates collected in ten Chilean cities between 2012 and 2017 (Table 6) were included in this study. All isolates were selected from the biorepository maintained by the National Institute of Public Health of Chile (ISP), Santiago, Chile. All isolates were cryo-preserved at −80 °C in glycerol (50% *v*/*v*) and trypticase broth (2:1). The ISP criteria were used to define HA-MRSA and CA-MRSA [20].

### 4.2. Antimicrobial Susceptibility Testing

The cefoxitin test (FOX, 30 µg) for methicillin resistance detection, D-test, iMLS_B_, cMLS_B,_ and MS phenotypes detection and antibiotics susceptibility determination, were performed by disk diffusion method on Mueller–Hinton agar following the CLSI recommendations and suggested breakpoints (2018) [27,28,29]. The antibiotics tested were erythromycin (ERY, 15 µg), clarithromycin (CLR, 15 µg), azithromycin (AZM, 15 µg), clindamycin (CLI, 2 µg), chloramphenicol (CHL, 30 µg), rifampicin (RIF, 5 µg), and trimethoprim/sulfamethoxazole (SXT, 25 µg).

The minimal inhibitory concentrations (MICs) of linezolid (LZD), vancomycin (VAN), and daptomycin (DAP) were determined using the broth microdilution method, according to CLSI guidelines and recommended breakpoints [28,29].

### 4.3. Characterization of MRSA Isolates

The presence of *mecA*, *pvl* in MRSA isolates, and the detection and characterization of the SCC*mec* element were performed by PCR-based protocols, as previously described [30,31,32]. Sequence types (ST) were obtained according to Opazo-Capurro et al. (2019), using the Pasteur’s scheme STs employing the bioinformatic tools available at the Center for Genomic Epidemiology (CGE) server (http://www.genomicepidemiology.org/, accessed on 13 March 2022) [33].

### 4.4. Molecular Detection of Antibiotic Resistance Genes 

The detection of genes involved in the MLS_B_ (*ermA*, *ermB and ermC*) and MS_B_ (*msrA*) phenotypes were screened by conventional PCR according to protocols and primers previously described [34] (Appendix A). 

### 4.5. Statistical Analyses

Pearson’s chi-squared test was used to determine associations between antibiotic resistance profiles, MLS_B_ resistance genes, and MRSA types (CA or HA-MRSA). This was achieved utilizing the IBM SPSS Statistics version 23.0 software (SPSS Inc, Chicago, IL, USA), establishing statistical significance at *p* < 0.05 [35].

## 5. Conclusions

In Chile, in isolates of HA-MRSA, there is an evident predominance of ST5-SCC*mec* I, a Chilean/Cordobes clone, characteristically multiresistant, which includes resistance to antibiotics from the MLS_B_ group; and susceptible to SXT and RIF. On the other hand, at the community level (CA-MRSA), there is an emergency of ST8-SCC*mec* IV, related to clone USA 300. Thus, microbiological surveillance of these isolates at the nosocomial level is required to verify whether the Chilean/Cordobes clone will be replaced by this community clone in Chile, and to monitor whether the latter will continue to increase its resistance to non-beta-lactam antibiotics, such as those of the MLS_B_ group.

## Figures and Tables

**Table 1 antibiotics-11-01000-t001:** Sequence types (ST) of methicillin-resistant *Staphylococcus aureus* strains isolated in Chile.

	ST 5	ST 8	ST 30	ST 105	ST 868	ST 923	ST 2802	Total
**HA-MRSA**	27	0	0	5	0	0	0	32
**CA-MRSA**	1	28	4	0	1	1	1	36

**Table 2 antibiotics-11-01000-t002:** Antibiotic resistance profiles among methicillin-resistant *Staphylococcus aureus* strains isolated in Chile.

Resistance Profiles					HA-MRSA *	CA-MRSA *
**CLI**	**ERY**	**AZM**	**CLR**	**CHL**	2 (6.3)	0
**CLI**	**ERY**	**AZM**	**CLR**		29 (90.6)	0
**CLI**	**ERY**	**AZM**	**CLR**	**RIF**	1 (3.1)	0
**ERY**	**AZM**	**CLR**			0	9 (25.0)
**ERY**	**AZM**				0	1 (2.8)
All susceptible					0	26 (72.2)

*** No. of isolates (percentage), CLI**: clindamycin, **ERY**: erythromycin, **AZM**: azithromycin, **CLR**: clarithromycin, **CHL**: chloramphenicol, **RIF**: rifampicin; **HA-MRSA:** Hospital-acquired methicillin-resistant *Staphylococcus aureus*; **CA-MRSA:** Community-acquired methicillin-resistant *Staphylococcus aureus*.

**Table 3 antibiotics-11-01000-t003:** Minimum-inhibitory concentration (μg/mL) of some antimicrobials against methicillin-resistant *Staphylococcus aureus* strains isolated in Chile.

Antimicrobials	MIC_50_	MIC_90_
**Linezolid**	2	2
**Vancomycin**	1	1
**Daptomycin**	0.25	0.25

**Table 4 antibiotics-11-01000-t004:** Antibiotic resistance, and presence of resistance genes in methicillin-resistant *Staphylococcus aureus* strains isolated in Chile.

	Percentage of Resistant Isolates to:						Percentage of Resistance Genes:			
	CLI	ERY	AZM	CLR	CHL	RIF	*ermA*	*ermB*	*ermC*	*msrA*
**HA-MRSA**	100	100	100	100	6.3	3.1	87.5	0	0	0
**CA-MRSA**	0	27.8	27.8	25	0	0	16.7	0	6.3	30.6

**CLI**: clindamycin, **ERY**: erythromycin, **AZM**: azithromycin, **CLR**: clarithromycin, **CHL**: chloramphenicol, **RIF**: rifampicin; **HA-MRSA:** Hospital-acquired methicillin-resistant *Staphylococcus aureus*; **CA-MRSA:** Community-acquired methicillin-resistant *Staphylococcus aureus*.

**Table 5 antibiotics-11-01000-t005:** Hospital-acquired methicillin-resistant *Staphylococcus aureus* isolates from different Chilean cities.

City	Number of Isolates
**Santiago**	15
**Rancagua**	2
**Talca**	1
**Concepción**	2
**Los Ángeles**	1
**Temuco**	3
**Osorno**	1
**Puerto Montt**	7
**Total**	**32**

**Table 6 antibiotics-11-01000-t006:** Community-acquired methicillin-resistant *Staphylococcus aureus* isolates from various Chilean cities.

City	Number of Isolates
**Valparaíso**	1
**Viña del Mar**	1
**Santiago**	14
**Rancagua**	2
**Talca**	1
**Concepción**	5
**Osorno**	1
**Los Ángeles**	1
**Temuco**	3
**Puerto Montt**	7
**Total**	**36**

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
