# Peer review of "Phenotypic and Genotypic Characterization of Macrolide, Lincosamide and Streptogramin B Resistance among Clinical Methicillin-Resistant Staphylococcus aureus Isolates in Chile"

_antibiotics, 2022, doi:10.3390/antibiotics11081000_

Round 1
Reviewer 1 Report
In this work, the authors report the resistance profile and genotype of 68 MRSA isolates in Chile. This is a rather short communication instead of a full research article. The manuscript is well written, but it is rather descriptive. Overall, it may be of use to very specialized clinical readers. There are some minor mistakes that should be corrected before the manuscript could be accepted:
Line 98. Instead of ST5, this should be ST105
Table 3, what is the breakpoint that you have taken as a reference to consider these isolates susceptible to the three antibiotics.
Line 105, why is this remarkable? In the discussion, the authors mention that this is consistent with previous results.
The authors should revise the Latin words in the whole document, many species names or gene names are not written in italics.
Author Response
Dear reviewer 1
Thank you very much for your corrections and suggestions. We have preceded to make the suggested changes in our manuscript:
1. Line 98. Instead of ST5, this should be ST105
It was corrected
2. Table 3, what is the breakpoint that you have taken as a reference to consider these isolates susceptible to the three antibiotics.
It was informed in material and method in the revised version of manuscript
3. Line 105, why is this remarkable? In the discussion, the authors mention that this is consistent with previous results.
This is remarkable, because these antibiotics still constitute a therapeutic alternative in clinical practice.
4. The authors should revise the Latin words in the whole document, many species names or gene names are not written in italics.
It was corrected throughout the body of the manuscript
Reviewer 2 Report
In their manuscript, Mario and co-authors assess the phenotypic and Genotypic Characterization of Macrolide, Lincosamide and Streptogramin B Resistance Among Clinical Methicillin-Resistant Staphylococcus aureus Isolates in Chile.
While this work is interesting, some claims from the title of the manuscript are more ambitious than the data and study design would justify.
-Moreover, the manuscript needs extensive English editing. The topic is very interesting but the authors have not presented the results in elaborative way!
-Aim of the study (line 83) and conclusions of the study are different.
-Methods are insufficient and are just supported by references.
-The study population is not clear at all.
-In results section, the analysis needs to be improved. I think more can be done with the data collected (and not easy to obtain).
- The conclusions are limited and will need reconsidering after the analysis is done in result section.
Please rewrite this sentence “Due to the above” line 56.
Line 57, as an important
Line 58, please rewrite this sentence
Line 90: what do you mean by SSCmec? You mean Staphylococcal Cassette Chromosome mec?
Line 92: How you confirm the absence of the pvl gene?
Line 97: what do you mean by this sentence? The MLST analyses of HA-MRSA showed that 27 (84.4%) isolates belonged to ST5 and 5 (15.6%) to ST5? 27 and 5 isolates both belong to ST5?
Line 102: You report that HA-MRSA were 32. But in this sentence they are not 32 but 35.
Thirty-one isolates of HA-MRSA (97%) were resistant to macrolides and to CLI. Moreover, 2 isolates (6%) were resistant to CHL and 1 isolate (3%) to RIF.
Line 118: What is SARM?
Line 130: Hard to understand this sentence!
Line 135: In Table 2, you have mentioned ermC as 6%, but in this sentence you mentioned 2%???
Line 163: fix this sentence! In this study were included? what was included?
Line 174: please clearly describe how you perform the Antimicrobial susceptibility testing? Disk diffusion? MIC (broth, agar)?
Line 184 and 188: It would be great if you provide a table for primers used in this study. Methods are an important section so better to write it in detail.
Author Response
Dear reviewer 2
Thank you very much for your corrections and suggestions. We have preceded to make the suggested changes in our manuscript:
1. While this work is interesting, some claims from the title of the manuscript are more ambitious than the data and study design would justify.
In the results, new information was incorporated that justifies the title assigned to this manuscript.
2. Moreover, the manuscript needs extensive English editing. The topic is very interesting but the authors have not presented the results in elaborative way!
We will request the suggested English editing services
3. Aim of the study (line 83) and conclusions of the study are different.
In the discussion (line 260, new version) the importance of performing the D test is summarized, but it is not a conclusion related to the objectives
4. Methods are insufficient and are just supported by references.
The description of the methods was expanded, keeping the references for greater detail of the methodology.
5. The study population is not clear at all.
This is explicitly described in the methodology. The origin of the strains is described in the text and in tables 5 and 6 (new version)
6. In results section, the analysis needs to be improved. I think more can be done with the data collected (and not easy to obtain).
The detail of the results was improved, incorporating more information and a new table (table 2).
7. - The conclusions are limited and will need reconsidering after the analysis is done in result section.
In the format of this type of article for the journal Antibiotics, the section is not mandatory
8.- Please rewrite this sentence “Due to the above” line 56.
It was corrected
9.- Line 57, as an important
It was corrected
10.-Line 58, please rewrite this sentence
It was corrected
11.-Line 90: what do you mean by SSCmec? You mean Staphylococcal Cassette Chromosome mec?
Yes, it was corrected
12.-Line 92: How you confirm the absence of the pvl gene?
This is explicitly described in the methodology. It was by PCR
13.-Line 97: what do you mean by this sentence? The MLST analyses of HA-MRSA showed that 27 (84.4%) isolates belonged to ST5 and 5 (15.6%) to ST5? 27 and 5 isolates both belong to ST5?
This was corrected. The MLST analyses of HA-MRSA showed that 27 (84.4%) isolates belonged to ST5 and 5 (15.6%) to ST105.
14.-Line 102: You report that HA-MRSA were 32. But in this sentence they are not 32 but 35.
Thirty-one isolates of HA-MRSA (97%) were resistant to macrolides and to CLI. Moreover, 2 isolates (6%) were resistant to CHL and 1 isolate (3%) to RIF.
This was corrected an changed to this text:
Thirty-one isolates(31/32) of HA-MRSA (97%) were resistant to macrolides and to CLI. Moreover, 2 isolates(2/32) (6%) were resistant to CHL and 1 isolate (1/2) (3%) to RIF.
16.-Line 118: What is SARM?
It was corrected and changed to MRSA
17.-Line 130: Hard to understand this sentence!
This was clarified. Lines 223-227 (revised manuscript)
18.-Line 135: In Table 2, you have mentioned ermC as 6%, but in this sentence you mentioned 2%???
It was corrected
19.-Line 163: fix this sentence! In this study were included? what was included?
This phrase explains how many and what type of MRSA isolates were included in the study. As stated in the text of the revised version:
In this study were included thirty-two non-repetitive HA-MRSA isolates recovered from eight Chilean cities between 2007 and 2017 (Table 5) and thirty-six CA-MRSA isolates collected in ten Chilean cities between 2012 and 2017 (Table 6)
20.-Line 174: please clearly describe how you perform the Antimicrobial susceptibility testing? Disk diffusion? MIC (broth, agar)?
The methodology was included in material and method section
20.-Line 184 and 188: It would be great if you provide a table for primers used in this study. Methods are an important section so better to write it in detail.
This was included in supplementary table
Round 2
Reviewer 2 Report
I did not find Lines 223-227 in the revised manuscript.
I am not satisfied with the answer to comment 7. The conclusions are limited and will need reconsidering after the analysis is done in the result section.
Author Response
Dear Reviewer 2,
Many thanks again for your suggestions. We have responded to your requests regarding our manuscript.
1) Moderate English changes required
The accepted version of the manuscript will be sent for English editing to the services offered by the publisher.
2) I did not find Lines 223-227 in the revised manuscript.
I have no idea why the numbers on those lines disappeared; however, the content (text) is correct and in the new version of the manuscript all lines are numbered
3) I am not satisfied with the answer to comment 7. The conclusions are limited and will need reconsidering after the analysis is done in the result section.
We have included a section of conclusions (No 5) (lines 218-225) and these are related to the objective of the work.